# Study of Residual Stress Using Phased Array Ultrasonics in Ti-6AL-4V Wire-Arc Additively Manufactured Components

**DOI:** 10.3390/s24196372

**Published:** 2024-10-01

**Authors:** Joseph Walker, Brandon Mills, Yashar Javadi, Charles MacLeod, Yongle Sun, Pradeeptta Kumar Taraphdar, Bilal Ahmad, Sundar Gurumurthy, Jialuo Ding, Fiona Sillars

**Affiliations:** 1Centre for Ultrasonic Engineering (CUE), Department of Electronic & Electrical Engineering (EEE), University of Strathclyde, Glasgow G1 1XQ, UK; brandon.mills@strath.ac.uk (B.M.); yashar.javadi@strath.ac.uk (Y.J.); charles.macleod@strath.ac.uk (C.M.); 2Department of Design, Manufacturing & Engineering Management (DMEM), University of Strathclyde, Glasgow G1 1XQ, UK; 3Welding and Additive Manufacturing Centre, Cranfield University, Milton Keynes MK43 0AL, UK; yongle.sun@cranfield.ac.uk (Y.S.); pradeeptta.taraphdar@cranfield.ac.uk (P.K.T.); sundar-gurumurthy@cranfield.ac.uk (S.G.); jialuo.ding@cranfield.ac.uk (J.D.); 4Centre for Manufacturing and Materials, College of Engineering, Environment and Science, Coventry University, Coventry CV1 5FB, UK; ac2333@coventry.ac.uk; 5Advanced Materials Research Laboratory (AMRL), Department of Mechanical & Aerospace Engineering (MAE), University of Strathclyde, Glasgow G1 1XQ, UK; fiona.sillars@strath.ac.uk

**Keywords:** phased array ultrasonics testing, wire-arc additive manufacturing, residual stress, contour method

## Abstract

This paper presents a study on residual stress measurement in wire-arc additively manufactured (WAAM) titanium samples using the non-destructive method of phased array ultrasonics. The contour method (CM) was used for the verification of the phased array ultrasonic results. This allowed for a comparison of measurement methods to understand the effects on the distribution of residual stress (RS) within Ti-6Al-4V samples and the effectiveness of measurement of residual stress using phased array ultrasonics. From the results of the experiments, the phased array ultrasonic data were found to be in good agreement with the CM results and displayed similar residual stress distributions in the samples. The results of the individual elements of the phased array were also compared and an improvement in accuracy was found. From per-element results, anomalies were found and could be mitigated with the ability to average the results by using phased array ultrasonics. Therefore, based on these results, there is a strong case for the benefits of using phased array ultrasonics as a method of residual stress measurement for WAAM Ti-6Al-4V components over other existing residual stress measurement techniques.

## 1. Introduction

Wire-arc additive manufacturing (WAAM) is a direct energy deposition process that has been increasing in popularity in the past few years as it can be used to create large metal components for different applications. Moreover, it also has a low equipment cost, a high deposition rate, and low material wastage, which also results in a lower impact on the environment [1].

There is great importance in understanding residual stress (RS) in WAAM Ti-6Al-4V as RS can play a role in causing and hastening damage within materials. This is caused by fatigue, stress-corrosion cracking, and RS-driven creep cracking, as well as by RS directly contributing to the driving force for fractures. Therefore, it is important to measure, evaluate, and attempt to mitigate RS, especially when considering components that are safety-critical [2].

The heat input required for arc sources during WAAM leads to a high buildup of RS in the component. RS is also associated with shrinkage during the cooling of the component and is highest along the direction of deposition [3]. Additive manufacturing, particularly WAAM, relies on the part being constructed layer by layer using melted material. However, as the material cools down, it then increases in temperature again once another layer is deposited on it, therefore leading to the material being heated and cooled several times. The unique nature of the thermodynamics of WAAM means that residual stresses (RSs) can form within the part in question, which can lead to distortions and microcracks, which in turn can lead to a failure of the manufactured part. Therefore, RS must be measured and mitigated as much as possible when manufacturing Ti-6Al-4V parts using WAAM.

For the last two decades, the contour method (CM) has been a well-utilised method for evaluating RS, particularly for material processing, where the evaluation of RS at a macro scale is necessary [4]. Originally introduced by Prime [5], CM is based on solid mechanics and can determine RS within an object using an experiment that is carried out by cutting a specimen into two pieces and then measuring the deformation caused by the RS distribution. The displacement data are then measured and used to create a finite element (FE) model of the sample, which computes the RS. This model takes into consideration the geometry and material stiffness of the sample, which provides a result unique to the specimen used. However, the main limitation of this method is that it is destructive and requires cutting the sample, rendering it non-functional.

Many non-destructive methods can be also used to measure strain and then stress, therefore being able to evaluate RS, with ultrasonic (US) testing being an important technique. This method of measuring RS was discussed by Noronha I and Weft [6]. Conventional US measurements are often carried out using single-element probes. The advantages of this method are its simplicity, affordability, and little required equipment with its ability to evaluate RS at different depths [7]. However, phased array ultrasonics offers an increased quality of inspection with a reduced inspection time. Also, phased array ultrasonics provides the ability to perform several inspections with one array and can provide instant images of the inspection [8].

Phased array ultrasonics has previously been underutilised within industries for RS measurements. Phased array ultrasonics testing (PAUT) RS measurement was first discussed by Javadi et al. [9]. However, there is a sizeable gap in research on ultrasonic phased arrays as a method for RS measurement.

### Ultrasonic Phased Arrays for Residual Stress Measurement

In their paper on the development of a PAUT-LCR system for RS measurement, Javadi et al. carried out a feasibility study for measuring residual stress in WAAM samples [9]. In their method, the Longitudinal Critically Refracted (LCR) ultrasonic technique was used as the goal was to measure stress in the bulk areas of the samples. However, this paper will expand on this feasibility study and use phased array ultrasonics to successfully measure RS in WAAM samples for the first time.

Although PAUT RS measurements were introduced by Javadi et al. [9], they did not go further than a feasibility study and results were not produced as only time-of-flight (ToF) variations caused by RS were discussed, not final RS measurements. While Javadi et al. [10] and Mills et al. [11] used similar principles for the measurement of bolt stress using phased array ultrasonics, they did not use them for RS measurement in WAAM components. The microstructures of the weld and WAAM, as well as the heat-affected zone (HAZ), bring more challenges in RS measurement than in the stress measurement of bolted connections. In this paper, phased array ultrasonics testing will be used for the RS measurement of WAAM components for the first time.

In the feasibility study carried out by Javadi et al. [9], they anticipated that the increase in the number of acoustic paths generated from their setup would therefore increase the measurement accuracy when compared to a traditional setup that generates two acoustic paths from three single-element transducers. As this was only a feasibility study, a comparison between the two methods was not made, but this estimated increased measurement accuracy was one of the focus areas of this study.

However, one of the main disadvantages of PAUT RS measurement, described by Javadi et al., is the issue of average data measurement. The average of the RS affected by the wave travel path, rather than point-based measurements, is measured. Therefore, when measuring RS at specific depths, it can include both bulk and surface stress data, and due to the rapid change in RS in WAAM components, the overall measurements are affected. Increasing the number of measurement frequencies in the method is one way to mitigate this issue. Another important factor to consider is the size of the probe. Larger arrays, due to the averaging issue, have problems measuring RS due to the sharp gradients within WAAM components. As a result, small-footprint arrays are the best alternative for WAAM RS measurements using PAUT.

In their paper on the development of the PAUT RS measurement system [9], Javadi et al. describe that another one of the main disadvantages of the ultrasonics method is that the ultrasonic wave is influenced by both the material texture as well as RS, making it difficult to differentiate between them. Also, in comparison to other RS measurement methods, only the average of RS is measurable using the ultrasonics method, limiting the method selectivity.

## 2. Theoretical Background

### 2.1. Residual Stress

Residual stress is a type of stress that is retained within a body of material when there are no external forces. RS can occur because of the incompatibilities of different regions of assembly, material, or components. A simple occurrence of an RS field is caused by an oversized sphere within a spherical cavity inside a large homogeneous body made of the same material. This can occur due to cooling a body that contains a sphere with a smaller coefficient of thermal expansion but with the same elastic constants [12].

Residual stresses are categorised into three main categories [13]:

**Type I**: Macro residual stresses that occur in the body of a component that is larger than the grain size of the material.

**Type II**: Micro stresses that vary depending on the scale of individual grains. These stresses typically occur in single-phase materials because of anisotropy in each grain.

**Type III**: Micro residual stresses that are within a grain; they occur as a result of dislocations and other crystalline defects. Both Type II and III stresses are micro stresses.

Examples of a Type I stress are the resultant stresses that are generated from the plastic bending of a bar and usually equilibrate over a length scale comparable to the structure in question. Type II stresses are typically produced due to differences in slip behaviour from grain to grain. Meanwhile, Type III stress is caused by point defects and by line defects due to doping with atoms with different amounts of radiation damage, which occurs at the atomic scale [12].

### 2.2. Contour Method

First discussed in 1957 by Hans Bueckner [14], Bueckner’s principle is an elastic superposition principle that states that if a cracked object is subjected to external loading and has forces also applied to the surface of the crack to close it, the forces need to be equivalent to the distribution of the stress in an uncracked body with the same dimensions/geometry that is also undergoing external loading on it.

This principle simplifies calculations and allows us to understand and calculate the intensity of stress and deformations using the original stresses on a cut plane. To do this, it is important to understand how RS or applied stress can lead to the growth of a crack that redistributes stress in a body. When a body is under stress, either applied or residual, the growth of a crack can occur, causing stress to release on the face of the crack and redistributing stress within the body. If the crack length increases, the redistributed stresses would then be released instead of the original residual stress. This is where Hans Bueckner devised a principle to correctly calculate everything and keep track of the redistribution of stress. This is the principle that is utilised within CM [15].

### 2.3. Conventional Ultrasonic Residual Stress

#### 2.3.1. Principles

The main idea behind US testing is the propagation of ultrasonic waves through a material to measure either the time of travel or change of intensity for a given distance, and sometimes both [16]. Two major factors affect the velocity of US within a material; these are the elasticity and density of the material. Young’s modulus represents the compressive and tensile stiffness of a material when a force is applied. The higher the value for Young’s modulus is, the higher the velocity of sound propagating through it will be. For example, steel has a high Young’s modulus value, and therefore, the velocity of sound travelling through it is typically high. On the other hand, the higher the density of the material is, the lower the velocity of sound travelling through it will be, so lead has a higher density than steel and, therefore, sound has a lower velocity when propagating through it. However, different waves have different velocities when travelling through materials. For instance, shear waves can travel through solids but do not have the same velocity as compression waves. This is caused by the modulus of rigidity of the solid, instead of Young’s modulus, which affects the velocity, which is lower than the modulus of elasticity [17].

#### 2.3.2. Snell’s Law

Snell’s law was discovered in 1621 and relates to the relationship between refraction and the angles of incidence for a light ray travelling through an interface of two isotropic media [18]. Snell’s law is represented by the formula in Equation (1).
(1)$$n_{1}sin\;\theta_{1}=n_{2}sin\;\theta_{2}$$

Equation (1) The formula for Snell’s law.

Here, *n*_1_ represents the refraction index of the first propagation media, *n*_2_ represents the refraction index of the second propagation media, *θ*_1_ represents the incident angle, and *θ*_2_ represents the refracted angle. Despite being mainly discussed in terms of light rays, Snell’s law also applies to ultrasound, and according to Snell’s law, when an ultrasonic wave crosses an interface between two materials, with both materials having different indices of refraction, reflected and refracted waves are both generated, and these are altered depending on the angle of incidence [19]. This is crucial for understanding how LCR waves can be created and used for measuring RS.

#### 2.3.3. LCR Waves

Ultrasonic waves can be split into four main types; these are transversal, longitudinal, Lamb and Rayleigh waves. Specifically looking at longitudinal waves, the particles oscillate in the direction of travel. When longitudinal waves interact with materials, several effects can alter the behaviour of the waves. For example, when a wave travels through two different materials, refraction, transmission, and wave reflection occur. During refraction, the angle of the wave propagation alters on the interface of the material. This is due to the difference in wave velocity for different materials, which is caused by different acoustic impedances. This phenomenon is expressed using Snell’s law. When the wave incident angle increases, so does the wave propagation angle in the material volume. Using steel as an example, the first critical angle level is around 28°, and so before this, the longitudinal wave is transformed into a transversal and partially to an LCR wave, which travels parallel to the surface of the material [20].

### 2.4. Phased Array Ultrasonics for Residual Stress Measurement

#### 2.4.1. Phased Array Ultrasonics for Defect Detection

PAUT employs constructive and destructive interference and utilises multiple elements compared to a typical single-element transducer, which uses only one. These groups of elements can be pulsed simultaneously, and the length of these active elements, or the total probe active length, is called the aperture [21]. PAUT does not always need to use multiple elements and can function with one aperture; however, some methods of PAUT can utilise multiple elements to provide advantages in testing.

One method of PAUT uses the beam-forming technique through the interference of the sound field in each element of the transducer. Compared to a typical ultrasonic probe, phased array probes are built up of numerous small transducer elements. This is how beams can be formed and steered. Each of these elements is individually excited by pulses, which causes a beam to be generated. Multiple beams, constructively and destructively, interfere to form a wavefront. The phased array equipment pulses each of these elements with time delays that are specified to form a wavefront that has been pre-calculated. For the instrument to receive, the reverse is performed. It receives the time delays that are pre-calculated and then sums the time-shifted signal to display them [22]. These pulses are controlled for transmission as well and the periods of the individual pulses for the delayed excitation of the individual elements are also set. These are set according to the size of each element, frequency, wave velocity, steered angle, or focal length for the inspection area in question. For a single signal, all the signals produced by the different elements are summed up into one, allowing for a sound wave to be enhanced in a specific direction or position [23].

#### 2.4.2. Phased Array Ultrasonics for RS Measurement

##### Principles

Ultrasonics RS measurement typically uses LCR waves, and the traditional setup for this method is extremely sensitive to material temperature, which can affect the accuracy of the results [24]. Therefore, instead of single-element transducers, ultrasonic phased array transducers can be used in a setup to improve the measurement accuracy of the LCR stress measurement setup. For this method, the two single-element transducers are replaced by two arrays with the required elements and MHz and two receivers are used to further improve accuracy. This tandem-catch setup minimises errors when measuring ToF caused by material texture effects and wave speed changes in the wedge or transducers or transmitter, triggering uncertainty [9].

##### Acoustoelasticity

The acoustoelastic constant is a dimensionless parameter that links the change in ToF and the stress or change in the velocity of the wave. This parameter increases because of tensile stress and decreases with compressive stress [25].

The acoustoelasticity law is used for measuring RS using the US method. This law states that the stress of a material affects the ToF of US waves. As previously mentioned, LCR waves have a higher sensitivity to stress when compared to other US wave types [26]. Based on this method, material stress can be determined using the ToF and the acoustoelastic coefficient measurements in both the material without stress and the material with either residual or applied stress. Using different US frequencies can also be used to penetrate a variety of materials with different thicknesses, which can be used to measure through-thickness RS. The transmitter and receiver transducers are placed into a wedge that can be moved over a material surface to extend the measurement coverage.

The acoustoelastic effect can only be shown by utilising extremely accurate measurements. For every MPa of stress applied to the material, only a 0.001% change in ultrasonic velocities is discovered [27]. Materials usually demonstrate substantial anisotropy and heterogeneity in their acoustic properties. The velocities of ultrasonic waves can be different in areas of the material that are free from stress. These limits of technological metals and the heterogeneity of acoustic properties are not entirely known [28].

The acoustoelasticity law states that material stress alters the time of flight (ToF) of US waves; therefore, Longitudinally Critically Refracted (LCR) waves are used due to their high sensitivity of stress when compared to other US waves.

Calculating US RS measurements relies on using both the ToF of the US waves and the acoustoelastic constant to find the RS in the sample. To do this, second-order elastic constants (Lamé constants) and third-order elastic constants are considered. When considering elasticity, the first Lamé constant is represented with *λ* and the second is represented by *μ*; this is the shear modulus of the material and is represented by Equation (1), where *μ* is the shear modulus, τ is the shear stress in the xy direction, and *γ* is the shear strain in the xy direction; these directions are later illustrated in Figure 6.
(2)$$\mu =\frac{\tau }{\gamma }$$

Equation (2): The shear modulus formula.

Along with the second-order elastic constants, the third-order elastic constants are also considered and are represented by A, B, and C. The three principal strains within the homogenous sample, categorised as *ω*_1_, *ω*_2_, and *ω*_3_, are combined and are represented by *θ* (a component of homogenous triaxial principal strains). With these considered, the ToF of longitudinal waves, parallel to the stress direction, can be related to strain (*ω*) with Equation (3).
(3)$${\rho_{1}V_{11}}^{2}=\alpha +\beta$$

Equation (3): Formula to calculate the ToF of longitudinal waves.

In this formula, *ρ*_1_ represents the density of the material, V_11_ represents the velocity of the longitudinal wave, *α* represents the terms related to the properties of the material, and *β* represents the terms related to the strain. With this formula, the rest of the terms can be put into the formula to create Equation (4).
(4)$${\rho_{1}V_{11}}^{2}=\lambda +2\mu +\left(2j+\lambda \right)\theta +(4k+4\lambda +10\mu )\omega_{1}$$

Equation (4): The complete formula for the relationship between the ToFs of longitudinal waves and material stress.

For this formula, *θ* represents the combined strain and ω_1_ represents the strain parallel to the stress. The relative sensitivity of velocity with strain then needs to be calculated to understand how the speed of the ultrasonic wave changes depending on the strain within the material. This is calculated using Equation (5):(5)$$\frac{L_{11}}{{\rho_{1}V_{11}}^{2}}=2+(v\mu \varepsilon )\frac{\omega_{1}}{\theta }$$

Equation (5): The formula for the relative sensitivity of velocity with strain.

*L*_11_ represents the acoustoelastic constant; in this equation, *L*_11_ is used to represent the fact that the acoustoelastic behaviour of the material is not dimensionless, unlike in other papers, such as that by Bray [29] where the authors use *L*_11_ as the dimensionless acoustoelastic constant. *ν* represents the value of Poisson’s ratio and *ε* represents the strain in the stress direction. To obtain the values required to calculate the acoustoelastic constant, the tensile testing of a Ti-6Al-4V sample needs to be carried out. This is conducted using both a tensile testing machine and ultrasonics testing equipment. For this testing, stress is increased in step intervals using the tensile testing machine while the ultrasonics equipment measures the ToF at each step [2]. Then, the stress variation in terms of ToF variations needs to be calculated to understand how changes in the ToF can relate to the variations in stress within the material. This is calculated using Equation (6):(6)$$d\sigma =\frac{\left(\frac{L_{11}}{{\rho_{1}V_{11}}^{2}}\right)\times \left(\frac{dt}{t_{1}}\right)}{\frac{\omega_{1}}{\theta }}$$

Equation (6): The equation for stress variation in terms of ToF variations (*dt*/*t*_1_).

Here, *dσ* represents stress variation and (*dt*/*t*_1_) represents ToF variation. Finally, the core formula for ultrasonic stress measurement can be used with a combination of the previous equations, which is shown in Equation (7):(7)$$\sigma =\frac{\left(\frac{L_{11}}{{\rho_{1}V_{11}}^{2}}\right)\times \left(T_{2}-T_{1}\right)}{T_{1}}$$

Equation (7): Core formula to calculate US RS measurements.

Here, *σ* represents the RS measurement, *T*_1_ represents the ToF of the US wave in the stress-free material, and *T*_2_ represents the ToF of the US wave in the material with applied stress or RS.

To find the acoustoelastic constant when considering all eight graphs, with each graph representing the ToF data of each array, Equation (7) is rearranged to create Equation (8):(8)$$L_{11}=\frac{\sigma T_{1}\rho_{1}V_{11}^{2}}{T_{2}-T_{1}}$$

Equation (8): Formula to calculate the acoustoelastic constant.

Equation (9) shows the formula used for the direct approach (*σ_D_*) for phased array ultrasonics RS measurements, where only direct acoustic paths from one transmitter to the receiver are considered.
(9)$$\sum \sigma_{D}=\frac{E}{L_{11}}\sum_{j=1}^{n}\left(\frac{T_{j}R_{j}\;WAAM-T_{j}R_{j}\;PARENT}{T_{j}R_{j}\;PARENT}\right)/n$$

Equation (9): Phased array ultrasonics—direct approach.

Here, *E* represents the elastic modulus, *n* represents the number of points, *T* represents the transmitter data, *R* represents the receiver data, *WAAM* represents the *WAAM* Ti-6Al-4V sample measurement data, and *PARENT* represents the parent (stress-free) material measurement data.

If we use the Full Matrix Capturing (*FMC*) (*σ_FMC_*) approach, Equation (10) can be used to generate results for all acoustic path possibilities instead. This allows for the acoustic path to transmit from any transmitter to any receiver and vice versa, generating more data than the direct approach.
(10)$$\sum \sigma_{FMC}=\frac{E}{L_{11}}\sum_{j=1}^{n}\sum_{k=1}^{n}\left(\frac{T_{j}R_{k}\;WAAM-T_{j}R_{k}\;PARENT}{T_{j}R_{k}\;PARENT}\right)/n^{2}$$

Equation (10): Phased array ultrasonics—FMC approach.

Figure 1 shows a schematic comparison of RS measurement using single-element transducers versus the phased array ultrasonics approach. In the single-element approach (Figure 1a), there are only one transmitter (T1) and one receiver (R1) transducer, so the ToF for the LCR wave is denoted as T1R1. T1R1 as measured in stressed material differs from T1R1 in unstressed material due to acoustoelasticity (Equation (7)). In the phased array ultrasonics approach (Figure 1b), there are two transmitter elements (T1 and T2) and two receiver elements (R1 and R2), creating a 2 × 2 matrix for both measurements in stressed and unstressed materials. None of the elements in this matrix are equal for the stressed and unstressed materials. If only T1R1 and T2R2 are considered, the approach is referred to as the direct method (Equation (9)). If all four elements of the matrix (i.e., T1R1, T2R1, T1R2, and T2R2) are considered, this approach is called the FMC method (Equation (10)).

## 3. Manufacturing of Sample

In this study, a plasma WAAM-deposited Ti-6Al-4V wall was utilised as the experimental sample, measuring a substrate of 250 mm × 60 mm × 7.3 mm in size. The samples were prepared using Ti-6Al-4V filler wire with a diameter of 1.2 mm. Ten layers were deposited with a wire feed speed of 2.2 m/min and a current of 180 A. During the deposition process, the plasma torch travel speed was adjusted, with the first two layers deposited at speeds of 4 mm/s and 4.5 mm/s, respectively, while a constant speed of 5 mm/s was kept for the subsequent layers. The layer height and width were measured to be 0.98 mm and 9.23 mm on average, respectively.

The Plasma Transferred Arc (PTA) was generated with pure argon serving as both the plasma and shielding gas. The flow rates for the plasma and shielding gas were set to be 0.8 L/min and 8 L/min, respectively. A local shielding device with a gas flow rate of 68 L/min was also integrated into the system.

It is important to supply a local shielding gas during this process to protect the area that is undergoing both solidification and melting to prevent oxidation. This often is in the form of a shroud that trails behind the beam and carries out the shielding. Argon is commonly used for shielding due to its density, which provides an improved efficiency of shielding [30]. The oxidation of the component can create slag inclusions and the evaporation of nitrogen during solidification can cause pores and nitrides, which can cause brittleness [31].

The experimental setup, depicted in Figure 2, positioned the plasma torch at a fixed distance of 8 mm from the substrate during deposition while the wire was inclined at a 25° angle to the travel direction. To maintain stability, the substrate was securely fastened using six clamps. The controlled movement of the PTA during deposition was facilitated by a six-axis KUKA robot operating in an alternating travel direction.

Figure 3 shows a schematic of the bottom of a WAAM Ti-6Al-4V sample and a photo of the physical sample itself and illustrates the different zones that were expected to have different RS measurements. Looking at the bottom of the sample, it can be visually seen that there was a distinct heat-affected zone (HAZ) in the centre of the sample. To help with choosing increments for the US RS measurement, different zones were specified, as shown in Figure 3. These related to the base metal area, the HAZ (Zone 1 and Zone 2), and the rest of the surface of the WAAM sample bottom. These zones were specified so that larger increments of measurements could be taken in the base metal area whilst smaller increments were measured in Zones 2 and 3. The centre of the HAZ of the WAAM sample was located at 0 mm, so *Zone 1* was measured to be from ~−5 mm to 5 mm, whilst *Zone 2* was measured at ~−10 mm to 10 mm at either side of the centre. Using these zones as a basis allows for a comparison with the different types of RS measurements to understand the accuracy of the RS measurements.

## 4. Materials and Methods

Figure 4 shows the process of the experiments that were carried out for the comparison of results in this paper. As shown in Figure 4, the overall process of experiments was split into two main measurement methods, which were CM and phased array ultrasonics.

For CM, the typical process was carried out with a separate WAAM Ti-6Al-4V sample being cut into two parts; the surface profile of the plane was measured to obtain displacement measurements, which were then processed to create an FE map of the RS across the surface of the cut sample. These results were what were compared with the phased array ultrasonics results. For the phased array ultrasonics results, the wedge angle for the LCR waves first needed to be determined by carrying out an angle/distance study. Then the measurements are split into two main parts. ToF measurements were collected from the Ti-6Al-4V sample. The acoustoelastic constant of the sample material was then found by carrying out tensile testing. Both these sets of data were post-processed and used to calculate the RS measurement results. Finally, both these results and CM results were compared.

## 5. Residual Stress Measurements

### 5.1. Contour Method

A WAAM Ti-6Al-4V sample was created, and this would be the basis for the sample that would be tested using both PAUT and CM, as shown in Figure 5. CM involves cutting a sample into two pieces to evaluate the RS and create a stress map of the sample. Figure 5 shows the cutting direction used for CM and the resulting cut sample. This was then used to produce the CM stress map and data for RS measurements.

The samples were cut using a Fanuc Robocut α-C600i wire electro-discharge machine equipped with a 0.25 mm diameter brass wire. The samples were clamped symmetrically, and the WEDM cut initiated from the top of the deposit, moving downward toward the bottom of the substrate. This process was carried out with low electrical power settings and a cutting speed of approximately 0.25 mm/min. The displacement profile of the cut sample surfaces was measured using a Zeiss Contura g2 coordinate measuring machine (CMM) using a 3 mm diameter drag probe. The distances from the perimeter and between the individual measurement points in both directions of the sample surface were set at 0.2 mm.

Post-processing of the displacement data from both cut surfaces of each sample involved aligning, cleaning, flattening, and smoothing using MATLAB analysis. Smoothing was achieved with a cubic spline knot spacing of 2.5 mm in both X and Y directions. An FE model of one cut half of the samples was developed, using the Abaqus software 2021, using an eight-node brick element (C3D8R). Constraints were applied to the model to prevent rigid body motion and the measured contour’s reverse was used as the displacement boundary condition. A linear elastic FE analysis was carried out to obtain the stress results with the following material properties: modulus of elasticity [E] = 113.8 GPa and Poisson’s ratio [ν] = 0.342. The resulting mesh and corresponding residual stress distribution from this FE analysis are shown in Figure 6.

### 5.2. PAUT

Before carrying out RS measurements on WAAM Ti-6Al-4V samples, it is first important to measure the acoustoelastic constant of the material, represented by L_11_ in Equation (4). To do this, an experiment was carried out involving iterative loading with ultrasonics measurements taken to find the acoustoelastic constant. The setup for this experiment using the Ti-6Al-4V sample is shown in Figure 7.

As shown in Figure 7, the Ti-6Al-4V was placed into an Instron tensile testing machine; clamps were then used to hold the PAUT transducers into place with the wedge on the Ti-6Al-4V sample. The setup for this PAUT method is shown in Figure 8. A PEAK Micropulse 6 phased array instrument was then connected to a laptop running LabVIEW to collect the ultrasound data at each interval. An initial test was carried out on the Ti-6Al-4V sample with 10KN increments of loading up to 240KN and unloaded back to 0KN. Finally, the main test for the Ti-6Al-4V sample was carried out at 2KN increments up to 250KN and unloaded back to 0KN. Using these data, acoustoelastic constants could be found that were able to be used alongside the PAUT data. As shown in Equation (7), RS is calculated using a combination of the calculated acoustoelastic constant for the sample material alongside the ToF of the US wave in the stress-free material and the ToF of the material with applied stress or RS.

A new measurement approach was developed for PAUT-LCR measurement that was improved over the traditional setup for ultrasonic LCR RS measurement. This setup is shown in Figure 8. For this setup, rather than using single-element transducers, which are typically used, two ultrasonic phased arrays are used. For the conventional LCR method, two receivers are used so the results are not impacted by the microstructure or material texture of the sample. This is possible due to the ability to generate two LCR wave paths that allow for the RS to be measured twice. However, when using the improved setup used for this paper, as shown in Figure 8, two 5 MHz arrays with eight elements each are used. With two eight-element arrays, the transmitter array generates eight ultrasonic waves, which are received by any of the eight elements from the receiver array. Therefore, if using FMC with this setup in the future, a potential 8 × 8 matrix of 64 LCR wave paths can be generated, increasing the accuracy of the results over that of the traditional ultrasonic LCR setup.

As shown in Figure 8, the PAUT-LCR RS setup comprises two eight-element transducer arrays; one is used as the transmitter and one is used as the receiver to measure the critically refracted waves. These two arrays are placed into an acrylic wedge that allows the ultrasonic waves to refract through it between the transmitter, material in question, and receiver array.

To design the wedge shown in Figure 8, the angle of the wedge was of great importance to ensure that the collection of data was accurate. Due to sound propagating differently in different materials, the type of wedge used is often different depending on the chosen material. Therefore, an experiment was carried out that involved the testing of the Snell’s-law wedge. This followed the logic previously discussed wherein Snell’s law relates to ultrasonic refraction to create LCR waves and uses the formula shown in Equation (1) to represent this. This would allow for an understanding of the gain required for the LCR peak to reach a certain threshold, which was measured in the study. Thus, if the angle was less suited, the LCR measurements would be inaccurate and require a higher gain to reach the required amplitude, and if the angle is more suited, a lower gain would be required.

An experiment was conducted for WAAM RS stress using the PAUT-LCR measurement method as shown in Figure 9. This test involved the measurement of RS in the WAAM Ti-6Al-4V samples using LCR waves from PAUT. The experimental setup for this is shown in Figure 10. For this test, a PEAK Micropulse 6 phased array instrument was used alongside the phased array LCR setup to measure residual stress in the samples. The results from this test could then be post-processed using the results from the acoustoelastic constant test previously mentioned.

Figure 10 shows the overall equipment setup for PAUT-LCR measurements of RS. The PAUT array setup was connected to a PEAK MP6 device, used to electronically drive the arrays, which connected to a laptop running LabVIEW to collect the measurements that could be post-processed later for measuring residual stress. As shown in Figure 8, two 5 MHz 8-element phased arrays were used, with one acting as a transmitter and the other acting as a receiver. Soundsafe industrial couplant was used due to its high acoustic transmission and was placed in between the arrays and the wedge they were placed into. The setup could be modified with different array setups depending on the material in question.

To collect more data, ToF measurements were taken at smaller intervals at the HAZ. As shown in Figure 10, a ruler was placed onto the bottom of the WAAM sample and the 20 mm point was aligned with the 0 mm measurement point. Using the ruler as a basis to understand where the scan lines were, RS measurements could be taken from −10 to 10 mm for the data points used in the results graph. This allowed for an easy comparison of the data between the RS measurement methods.

Figure 11 shows an example signal with a close-up of the LCR wave and the zero crossing from the ToF measurements. To create the final RS measurement results, the zero crossing was taken from each of the eight signals. These zero-crossing values could then be input alongside the acoustoelastic constant data to produce the RS measurements.

## 6. Results and Discussion

As described previously, RS measurements were collected using the CM and PAUT methods as seen in the FE model in Figure 12, which was created using the CM. Scan lines were determined that can be used to measure the RS in specific areas of the sample using the phased array ultrasonics method. As the purpose of this study was to demonstrate the ability for WAAM RS measurement using the phased array setup, the scan area for the comparison of data for these results was focused on the Lc scan line. In the figure showing the FE model, the Lc scan line at the bottom of the sample can be seen to contain a high amount of RS. Therefore, by choosing this area, an easy comparison can be made between the phased array and CM results. The Lc scan line was also chosen as the penetration depth of the 5MHz transducers used was measured to be 1.5 mm, which was equivalent to the Lc line in the CM results, as that line also showed the data at 1.5 mm.

Towards the centre of the bottom of the sample, the highest amount of RS was present where RS remained high but was slightly lower within *Zone 2*.

As seen in the CM results shown in Figure 13, the stress gradually increased to a peak of 400 MPa at 30 mm along the substrate. The stress then began a gradual decline to the lowest point of −150 MPa at 55 mm along the substrate and then increased again to 100 MPa between 55 and 60 mm.

There were a few anomalies found in the CM data. As seen in Figure 13, between 0 mm and 1 mm there was a slight spike in stress from −200 MPa to −150 MPa, which then decreased again to −200 MPa before increasing gradually along the substrate as mentioned before. A similar occurrence happened between 45 and 60 mm, where stress increased and decreased several times towards the end of the substrate.

Another factor that stood out from the CM results was the overall shape of the data. We would expect, from RS measurements carried out on the WAAM sample, that the shape of the graph would be symmetric due to the overall manufactured shape of the sample. There were a few possibilities to explain this unexpected graph shape. One was that the shapes of the WAAM samples that were being tested had a slight bow to them and were not completely flat; a visual representation of this bowed shape can be seen in the cross-sectional geometry in Figure 14, created using a laser scan where the bottom of the WAAM sample can be seen to not be flat. The slight distortion during the production process was due to the stress on the sample during the manufacturing process, and this could have affected the stress values as the distortion of the sample relieved some stress. It is believed that this distortion did not interfere with the ToF measurement for two reasons. (I) the distortion resembled angular shrinkage rather than buckling, which could have affected surface flatness. This was studied by Satarri-Far and Javadi [32], who found that distortion on welded pipers did not affect the final RS measurements when comparing the finite element results and the experimental data. It should be noted that surface flatness can influence ToF measurement, especially if the width of the PAUT wedge is too wide to account for surface deformations. (II) The width of the PAUT wedge was 8 mm, which made it significantly narrower than any potential surface deformation on this sample.

Efforts were carried out to prevent the distortion of the sample as much as possible. The substrate was flat before deposition and six clamps were used with each side of the substrate having three clamps placed evenly in the longitudinal direction, creating a symmetrical clamping condition.

Shown in Table 1 are the results from the Snell’s-law experiment to find the correct angle for the wedge for use with Ti-6Al-4V when collecting phase array ultrasonics measurements. From these results, the average difference in gain from the Snell calculated angle wedge could be compared against the wedge angle for titanium. These wedge angles were tested between 22.2 and 23.2 degrees as these are the approximate angles for the first critical angle level when using LCR waves with titanium alloys as a material. Therefore, by measuring within this area, a specific wedge angle degree could be found that was most suitable for the RS measurements.

From these results, the most important thing to consider is that the lower the average difference in gain is, the more suitable the wedge is for titanium, as less gain is required to reach the required amplitude necessary to collect the ToF measurements, ensuring that they are accurate.

As seen in Table 1, the lowest values were found at 22.6 and 22.7 degrees, and therefore, both wedge angles were suitable for ultrasonic phased array testing in Ti-6Al-4V samples; ultimately, a 22.7-degree wedge was chosen for this paper.

To measure the acoustoelastic constant, tensile testing was carried out, with stress increments in both loading and unloading modes, and the results are shown in Figure 15 and Figure 16, respectively. Since two eight-element arrays were used, the results have been presented per element. For example, Element 4 shows the ToF is measured for the acoustic path between Element 4 of the transmitter array and Element 4 of the receiver array. The vertical axis shows the Exdt/t0, so the slope of each of these graphs represents the acoustoelastic constant, L_11_, in Equation (7). However, the final acoustoelastic constant can be calculated by considering all 16 graphs presented in Figure 15 and Figure 16 using Equation (8). Since it increases with an increasing stress and decreases with a decreasing stress, the loading and unloading graphs are ascending and descending, respectively. However, several points do not follow this general trend. For instance, in Element 6 in Figure 15, there is a spike at around 350 MPa, followed by an unexpected decrease at the next point. This issue can be attributed to one of the ultrasonic RS-measurement-system errors discussed by Javadi et al. [9] such as in the couplant film thickness, a triggering error, or a tensile-testing machine error. However, it is important to note that the acoustoelastic constant would have been measured using all 200 (the number of data points in each graph) × 16 data points collected during the tensile testing process, which would have helped minimise the system error. This is one of the main advantages of a phased array over a single-element ultrasonic system, where only one of these graphs (instead of sixteen) would have been generated. Although a single averaged graph can be generated by using the ultrasonic phased array system, by generating 16 individual graphs, any anomalies/errors can be spotted and considered for the final RS results. However, with a single-element ultrasonic system, the singular graph could potentially contain errors without the ability to compare them to the average trend of the results to spot them.

Using the acoustoelastic constants, the final phased-array WAAM RS measurements could be created. Once the RS measurements were collected for each element of the phased array transducer, these measurements could then be averaged to produce a single set of RS measurements. However, it was important to ensure the accuracy of the results and to see whether there were any specific outliers or incorrect measurements from any elements. To find these outliers, the variance between the RS measurements of each element and the averaged results were calculated and are shown in Figure 17. Any drastic difference between the RS measurements of each element and the averaged results could be visually seen and removed to improve the accuracy of the improved averaged results. To ensure the consistency of the chosen outliers, any measurements with a variance of 5 or above were identified as outliers as measurements above this variance were found to be commonly outside the normal RS measurements, as can be seen at −8 mm on Element 3. As can be seen in Figure 17, Element 2 most notably had the most outliers, with five measurements identified as outliers, which shows that this element could have been problematic. A variance of above 40 was also calculated at 8 mm for Element 5, which would have greatly affected the final results if not removed. Element 4 also had numerous outliers, with three identified. These outliers for the individual element results could be attributed to the normal ultrasonic stress measurement errors that were reported by Javadi et al. [9] when using the conventional single-element technique for LCR RS measurements.

In Figure 18, the stress distribution measurements per element are shown. The results show eight separate elements that have all been used to measure the stress distribution of the same WAAM sample. These results present the measured residual stress at a depth of 1.5 mm from the bottom of the sample, with a depth of 1.5 mm being the most effective area for the measurement of the RS in the sample.

These results have been presented displaying the RS measurements for each element. However, there are a few outliers in these results. The points displayed in red are the RS measurements that were identified as outliers when calculating the variance and using Figure 17. When considering the outliers and overall shape of the results of each element, Element 2 is most noteworthy as it displayed a very different RS distribution when compared to the rest of the elements. Element 3 and Element 7 seemed to show the most accurate RS measurements with few outliers and similar RS distributions. In all the measurements, the overall RS distribution was as expected as represented by the zones illustrated in Figure 3. Outside of the HAZ (−10 to 10 mm), a low RS was measured at ~0 to 20 MPa, and when measured at the start of the HAZ, the RS spiked and was measured at ~200 to 300 MPa in all elements, which occured on both sides of the HAZ. In all elements, peak RS was measured at 0 mm, which had been expected. However, within the HAZ, all the elements displayed a similar drastic drop in the RS at both sides of the HAZ from ~0 to −150 MPa.

These drops in the RS and other outliers could have been due to similar issues during the measurement process mentioned previously for the acoustoelastic results discussed by Javadi et al. [9] such as the triggering of errors, the couplant film thickness, or small angle variances with the array wedge. These factors are difficult to control during an experimental process, but future work can be controlled to mitigate their impacts. For example, one may conduct a phased array experiment using arrays with higher numbers of elements to have better control and reassurance in identifying the potential outliers.

To create the final averaged results for the RS of the WAAM Ti-6Al-4V sample using the phased array ultrasonics method, ToF measurements were used alongside the calculated acoustoelastic constant, and these results are shown in Figure 19. These measurements have been shown at a depth of 1.5 mm due to the effective depth of the phased array ultrasonics equipment. These results are averages of the per-element results taken from each of the eight elements in the phased array transducers. The red line represents the actual stress measured, and includes the outliers that were previously identified in the per-element results in Figure 18, and the black line removes these outliers. As can be seen, the removal of these outliers results in more symmetrical results in comparison and represents an improvement in accuracy.

These results follow a similar trend to the per-element results, as expected, with lower RS measurements outside of both Zone 1 and Zone 2, where less heat is present during the manufacturing of a sample. However, the RS increased from −10 mm until Zone 1, where the sample was most affected by heat, and there was a vast increase in the RS. From −5 mm, RS increased to ~220 MPa and peaked at 0 mm to ~310 MPa, where heat was most prevalent during fabrication. As discussed previously for the per-element results, there was a drop in RS measured on both sides of the HAZ at both between −7 and −4 mm, where RS dropped to ~60 MPa, and between 3 and 5 mm, where RS dropped to -27 MPa. Another significant drop in RS was measured at −2 mm at ~−80 MPa. These drops in RS in the HAZ could sometimes be expected within this area of the sample or could be attributed to the issues during the measurement process discussed for the per-element results. Also, the overall trend of the data was not perfectly symmetric. However, this had been expected, as the shape of the sample was slightly distorted, as shown in Figure 14.

To create the averaged phased array results from the per-element measurements, it was important to consider the outliers found in the per-element measurements. To average correctly, we needed to divide the sum of the black points (measured RS) by the number of points as shown in Equation (9). However, instead of simply considering the number of elements used, we needed to consider the number of elements minus the red points shown in the per-element results (outliers in HAZ) for the specific point being considered. For example, some of the measurements only had a single outlier/red point, such as Elements (3), (5), and (7), so we summed the black points and divided the resulting value by seven.

Doing this mitigated the outliers affecting the RS measurement for the averaged results and was an advantage of using phased array ultrasonics to average the measurements rather than relying on single-element measurements. As previously mentioned in Section 1, Javadi et al. [9] anticipated that the increased acoustic paths from the LCR approach with phased arrays would increase the measurement accuracy when compared to using just two acoustic paths from the three single-element transducers in the traditional method. Although not used in this study, this phased array ultrasonic setup can also be utilised with FMC to create many more data. By using the two eight-element arrays for this paper, a potential 8 × 8 matrix of 64 LCR wave paths could be generated, increasing measurement accuracy. Also, with the ability to produce eight sets of RS measurements, the amount of data was increased, and if arrays with a higher number of elements were used, this same setup could be used to produce even more data. This is especially important if problematic elements, such as Element 2, are identified, which would have affected the accuracy of the averaged results. With a higher-element array, the impact of these problematic elements would be reduced.

## 7. Comparison of Results

When looking at the CM and the per-element results, the overall shape of the graph was more symmetrical to the RS measurements when considering the CM FE model depicted in Figure 6. As shown in the FE model, RS was low in areas outside of the HAZ whilst towards the centre of the sample where the HAZ was present, from −5 to 5 mm, the RS was much higher, which matched the RS measurements using ultrasonics testing.

Figure 20 shows a comparison between the CM results and the phased array ultrasonic results. Comparing the averaged phased array ultrasonics results in Figure 19 with the CM results in Figure 13, shown in Figure 19, there was an improvement in the overall measurement of the RS distribution within the sample due to the ability to remove known anomalies, which was not possible with the CM. The CM results showed a measurement of −200 MPa on one end of the sample at 0 mm and 100 MPa on the other end of the sample, showing that there was a large disparity in the measured RS outside of the HAZ. On the other hand, the phased array ultrasonics results showed a more symmetrical distribution of stress with RS measured similarly on both ends of the HAZ at ~20 MPa on each side. Both the averaged phased array and the CM results measured a similar peak RS at 0 mm, with the CM measuring it at ~420 MPa and the phased array measuring it at ~310 MPa.

Within the HAZ, the distributions of stress were similar in both results, with a slightly lower RS on one side of the area. These low RS measurements were flipped for the averaged results; however, this was due to the CM being scanned the opposite way. Although both results displayed a slightly asymmetric trend due to the shape of the WAAM sample, the non-symmetry was much more apparent in the CM results.

Set side by side, the phased array ultrasonics results agree better with the CM results compared with the per-element results. As previously discussed, the CM results represent a clear increase in RS at the HAZ between −10 and 10 mm with the highest RS expected to be between −5 and 5 mm. When comparing the shapes of the RS distributions, the phased array ultrasonics results align well with this increase in RS, which can be presumed to reaffirm the accuracy of the results.

Comparing the per-element results with the averaged phased array ultrasonics results, the latter shows a noticeable improvement in the accuracy of the overall RS measurements. By having the ability to compare the RS measurement of each element with the averaged results, outliers can be identified and removed to produce averaged phased array ultrasonics results with improved accuracy. Problematic elements, such as Element 2 in Figure 17 and Figure 18, can also be recognised when one is able to compare the RS measurements of the element with an averaged set of data.

Although not carried out in this study, FMC can also potentially be utilised with this phased array ultrasonics testing method, and with the two eight-element phased arrays used for this study, a possible 64 results can be produced. When capturing the acoustoelastic data, a large number can be taken to generate 200 (data points) × 16 (data points collected during tensile testing to reduce the effect of system error) sets of measurements. Therefore, eight graphs can be averaged, with each one of them being generated through the acoustoelastic constant, which itself has been generated using 200 × 16 sets of data. By averaging the eight individual element results with a much larger dataset available compared to a measurement with a single-element transducer, a single graph can be produced, which can improve accuracy and is a notable advantage of using phased array ultrasonics testing.

## 8. Conclusions

In this paper, a study on a phased array ultrasonic system for the RS measurement of WAAM Ti-6Al-4V samples, compared with the CM, was carried out. Tensile testing was also conducted for both loading and unloading with 200 load increments to measure the acoustoelastic coefficients of WAAM Ti-6Al-4V as the parent material for calibration to use for the discussed ultrasonics method.

From the results, the following can be concluded:By carrying out an initial Snell’s-law experiment to determine the wedge angle for the LCR method, a 22.7-degree angle was found to be the optimised wedge angle for Ti-6Al-4V.The results of phased array ultrasonics for the RS measurement of WAAM Ti-6Al-4V not only demonstrate the viability of this new method but also show its advantages over other common RS measurement methods.When comparing the per-element results and the averaged phased array results, there was a clear improvement in the measurements. Using the final averaged results, several outliers that went against the averaged results were able to be identified in the per-element results. These clear outliers could be removed as they affected the final averaged results of all eight elements.Phased array ultrasonics testing allows for more data to be produced compared to both the CM and using single-element ultrasonic transducers. For this study, eight different sets of results were created as the phased-array-ultrasonics direct approach was used with eight arrays to demonstrate its capability. However, although this was not used for this study, this setup has the potential to be used with FMC, and future work on this method can demonstrate the feasibility for it to be used with FMC. With the method described in this paper, where eight elements are used alongside FMC, 64 results can be produced along with acoustoelastic data with 200 load increments: eight data points during the loading and eight data points during unloading throughout the tensile testing, which can reduce system error effects. Therefore, by having a much larger dataset, errors can be mitigated.Quantitatively, the results did not agree fully. For instance, for the RS measured at the centre of the phased array, the ultrasonics results indicated ~310 MPa, but the CM was measured at ~420 MPa. Qualitatively, the phased array ultrasonics measurements agreed with the CM results and displayed a similar RS distribution. Both results showed similar symmetries and measured higher RS in the centre of the WAAM Ti-6Al-4V at the HAZ.

Based on the results discussed in this paper and the numerous advantages of the method over other RS measurement methods, it is strongly recommended to adopt phased array ultrasonics as a WAAM Ti-6Al-4V RS measurement method.

## Figures and Tables

**Figure 1 sensors-24-06372-f001:**
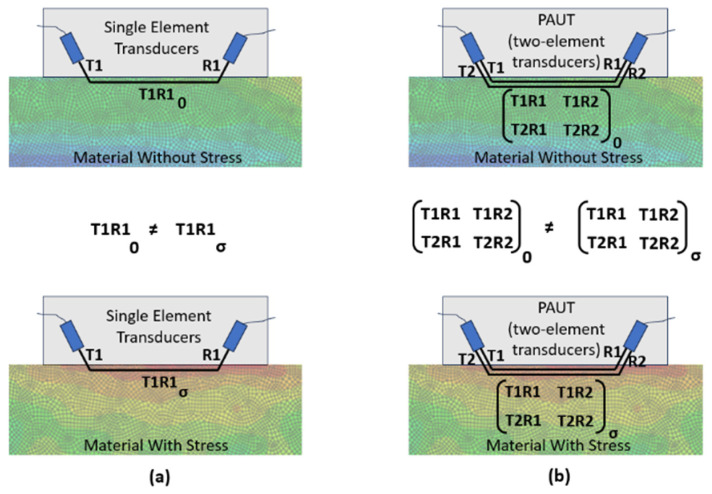
Schematic of residual stress measurement using (**a**) single-element transducers versus (**b**) phased array ultrasonics approach.

**Figure 2 sensors-24-06372-f002:**
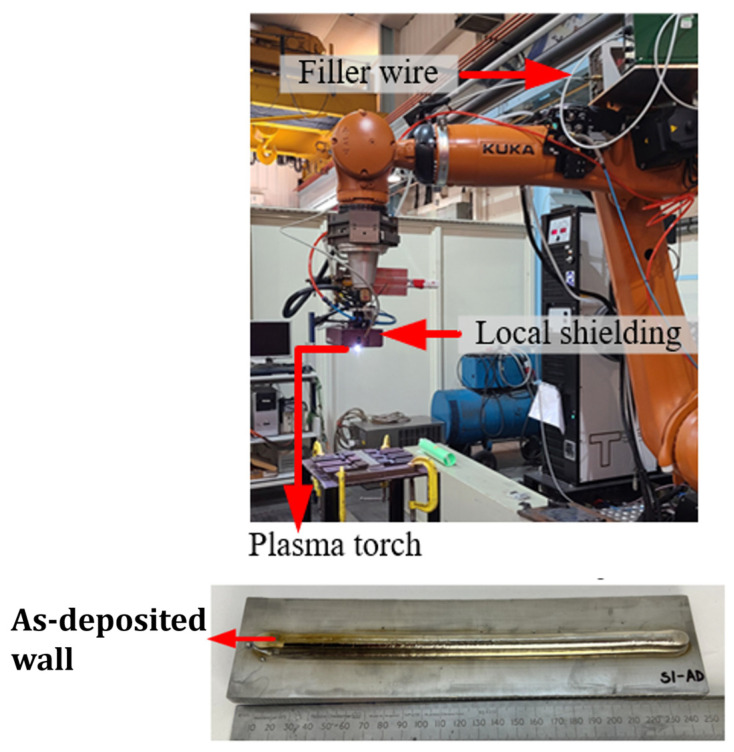
Experimental setup for the manufacturing of the Ti-6Al-4V samples.

**Figure 3 sensors-24-06372-f003:**
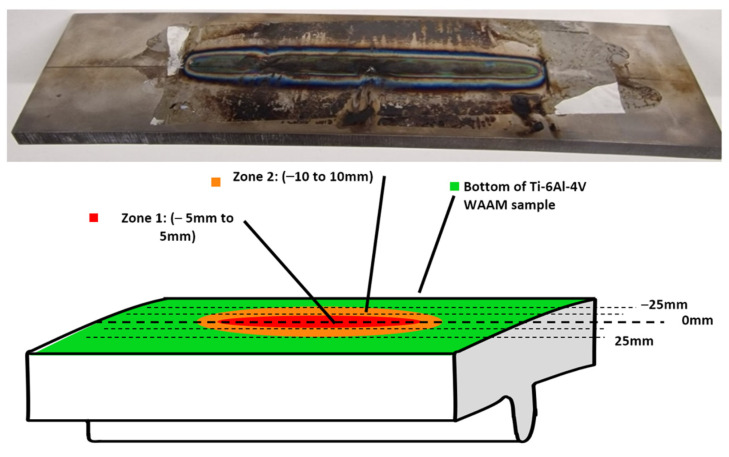
Schematic of the bottom of the WAAM Ti-6Al-4V sample showing an estimation of the HAZ.

**Figure 4 sensors-24-06372-f004:**
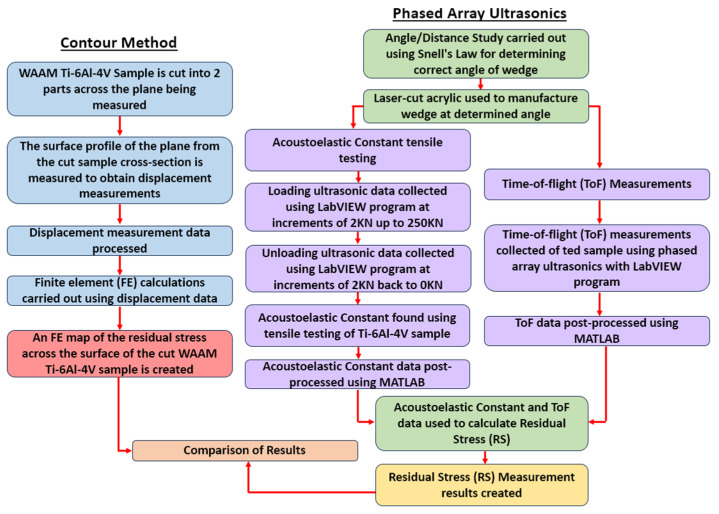
Flow chart of the process of carrying out the experiments.

**Figure 5 sensors-24-06372-f005:**
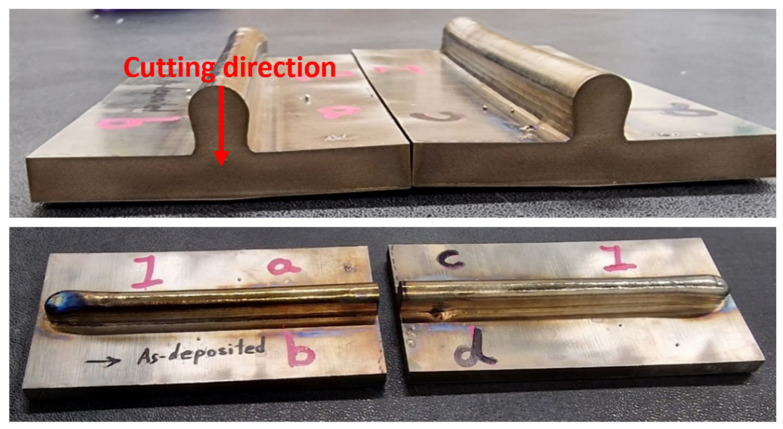
The cutting direction used for cutting the Ti-6Al-4V sample for CM and the resulting cut sample.

**Figure 6 sensors-24-06372-f006:**
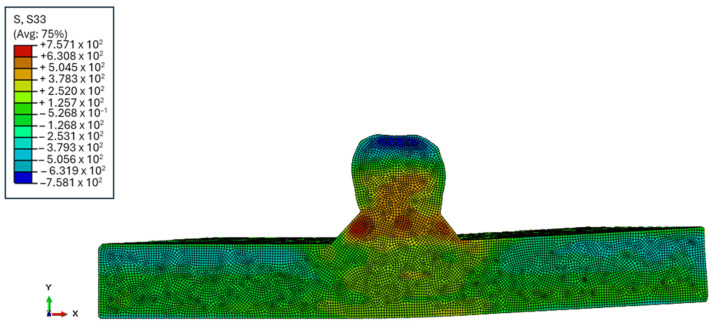
FE analysis mesh and back-calculated RS (in MPa) using CM.

**Figure 7 sensors-24-06372-f007:**
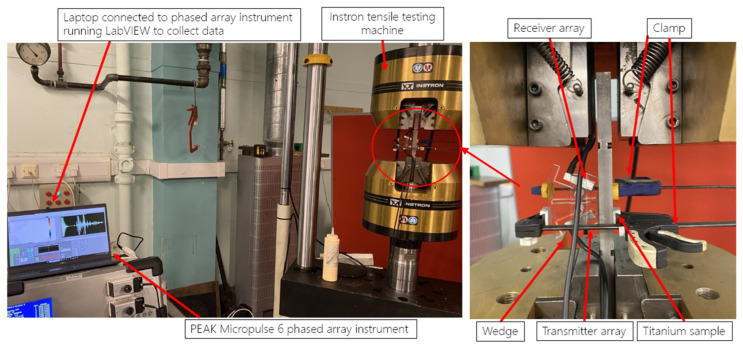
Experimental setup for finding the acoustoelastic constant of a Ti-6Al-4V sample.

**Figure 8 sensors-24-06372-f008:**
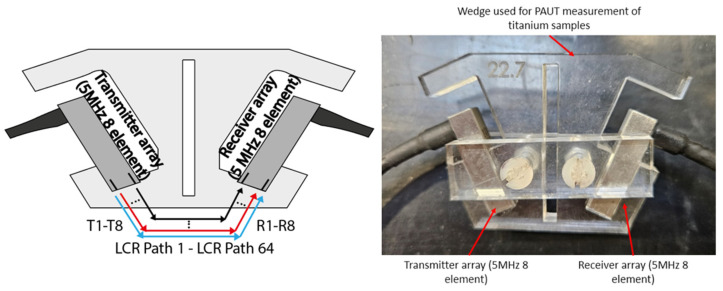
PAUT-LCR RS measurement approach setup.

**Figure 9 sensors-24-06372-f009:**
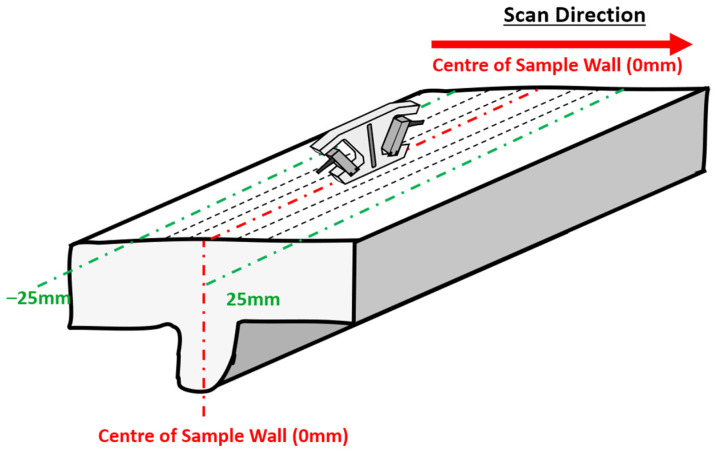
The scan path and direction when carrying out PAUT on the Ti-6Al-4V sample.

**Figure 10 sensors-24-06372-f010:**
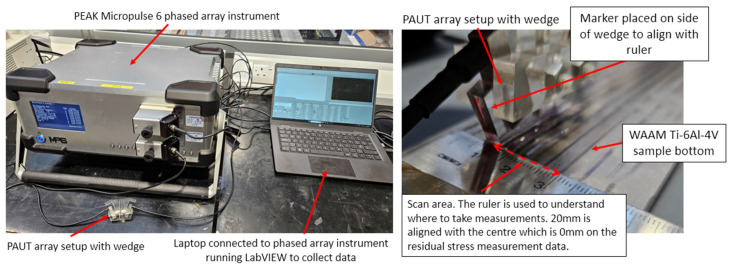
PAUT-LCR RS measurement setup showing the PEAK MP6 using two 8-element 5MHz arrays acting as transmitter and receiver, connected to the laptop for ToF measurements, which were then calculated with acoustoelastic measurements for RS measurements.

**Figure 11 sensors-24-06372-f011:**
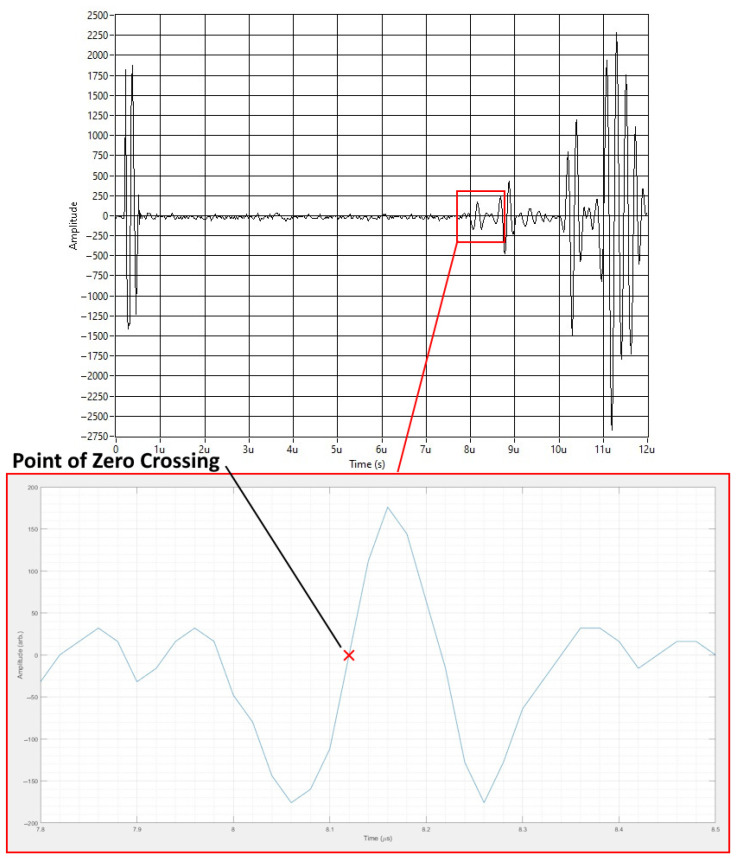
An example signal showing how the zero crossing is found from the LCR wave.

**Figure 12 sensors-24-06372-f012:**
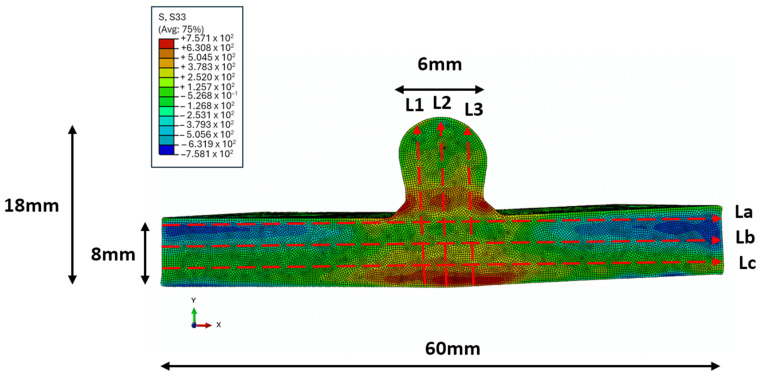
CM FE model with proposed scan lines for RS measurement in Ti-6Al-4V WAAM samples.

**Figure 13 sensors-24-06372-f013:**
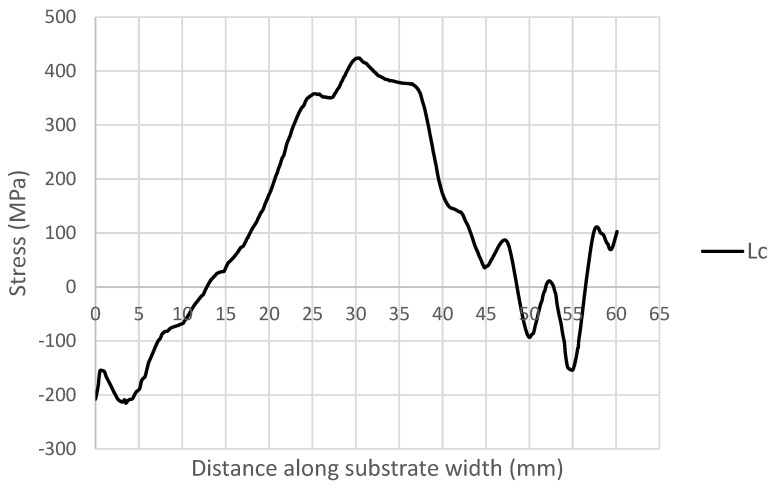
CM RS measurement results.

**Figure 14 sensors-24-06372-f014:**
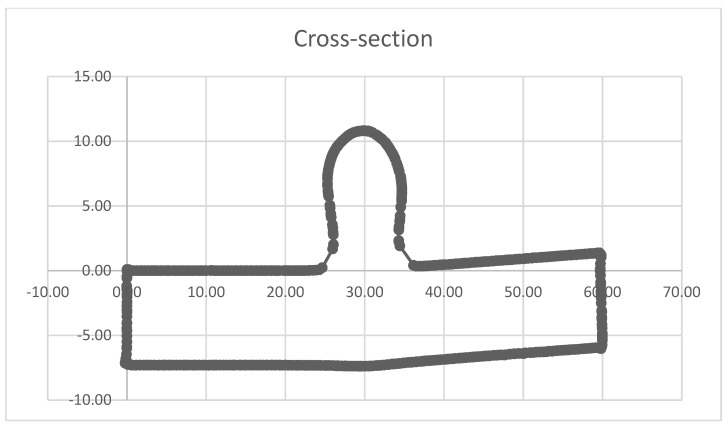
Cross-section of the WAAM sample.

**Figure 15 sensors-24-06372-f015:**
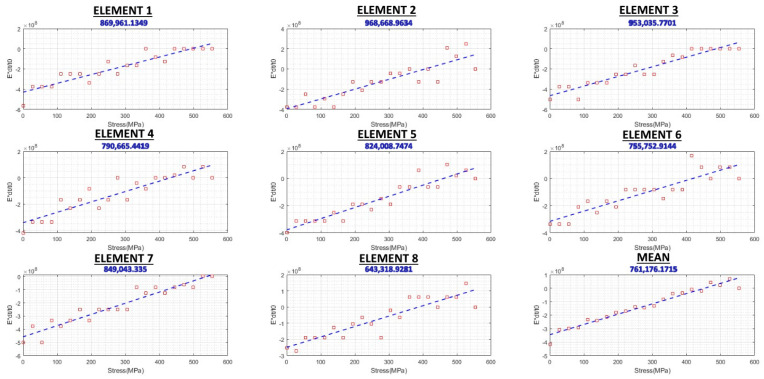
Loading/ascending acoustoelastic constant graphs.

**Figure 16 sensors-24-06372-f016:**
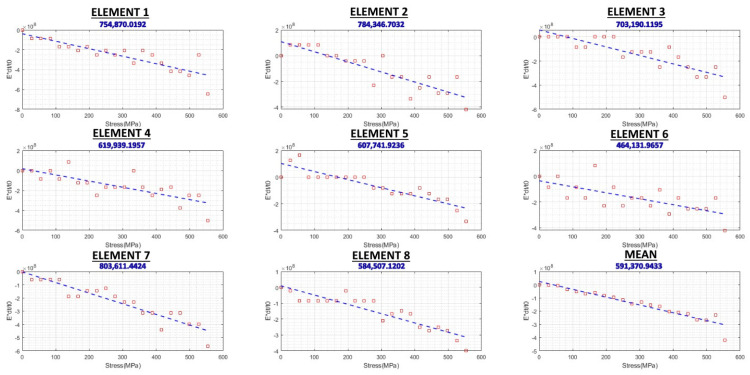
Unloading/descending acoustoelastic constant graphs.

**Figure 17 sensors-24-06372-f017:**
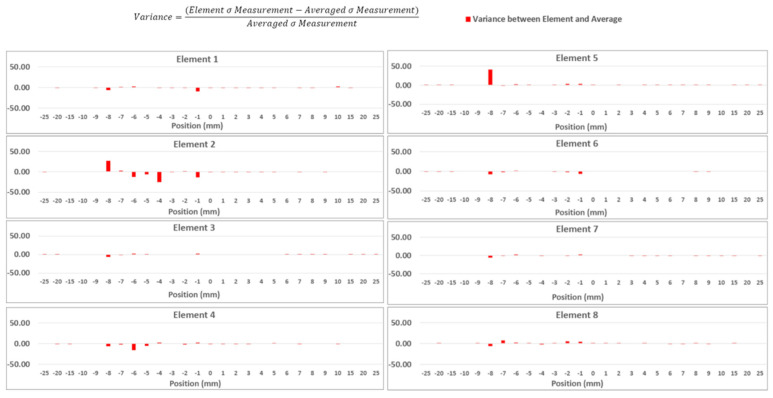
Calculated variance between each element’s RS measurement and averaged results.

**Figure 18 sensors-24-06372-f018:**
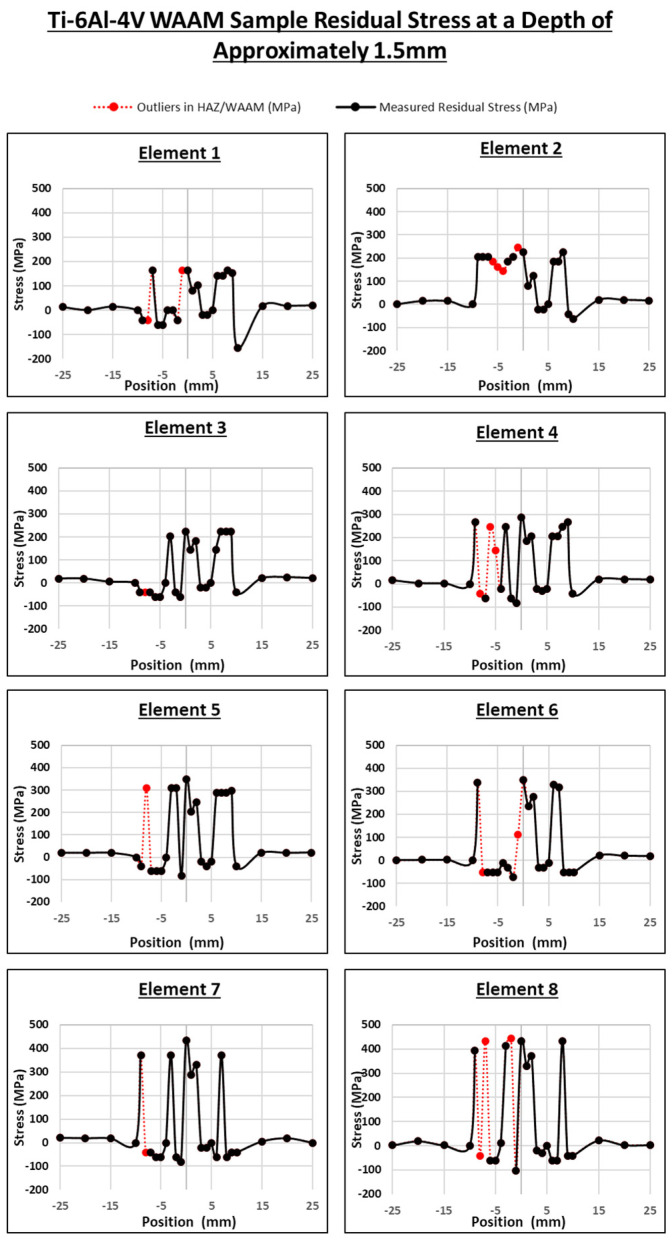
Per-element stress distribution across WAAM sample results.

**Figure 19 sensors-24-06372-f019:**
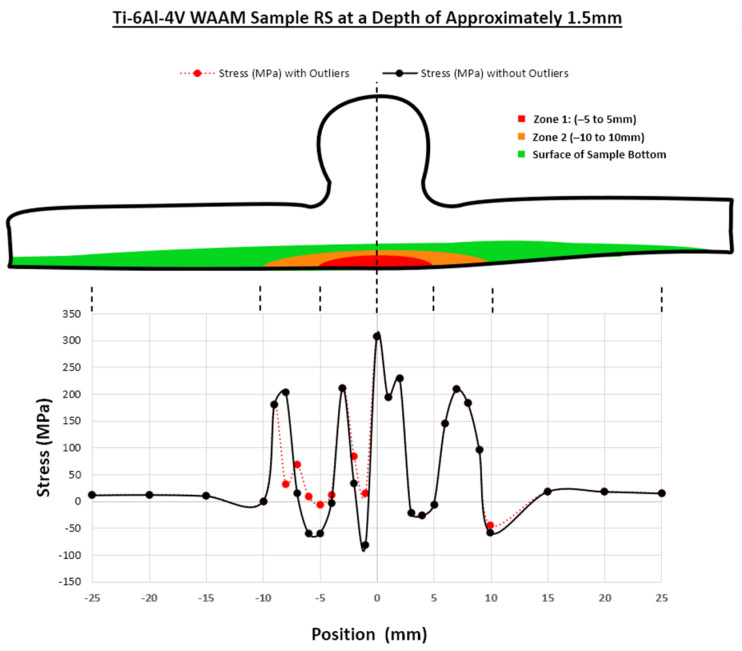
Results of the RS measurement using the phased array ultrasonics method.

**Figure 20 sensors-24-06372-f020:**
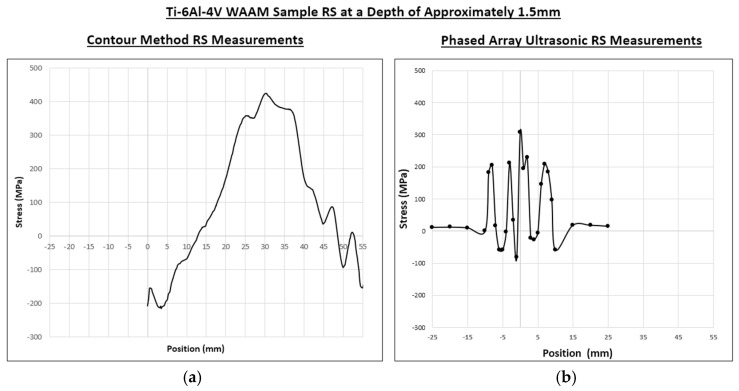
(**a**) The contour method RS measurements and (**b**) averaged phased array ultrasonics RS measurements.

**Table 1 sensors-24-06372-t001:** Average difference in gain relative to Snell-calculated angle wedge vs. wedge angle for titanium.

Wedge Angle (Deg)	Average Difference in Gain from Snell Calculated Angle vs. Wedge Angle for Titanium (dB)
22.2	14
22.3	8.5
22.4	1
22.5	5.5
22.6	0
22.7	0
22.8	11
22.9	4.5
23	4
23.1	14.5
23.2	11.5

## Data Availability

The data presented in this study are available on request from the corresponding author.

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
