# Peer review of "Study of Residual Stress Using Phased Array Ultrasonics in Ti-6AL-4V Wire-Arc Additively Manufactured Components"

_sensors, 2024, doi:10.3390/s24196372_

Round 1
Reviewer 1 Report
Comments and Suggestions for Authors
This paper focuses on the RS measurement of WAAM Ti-6Al-4V samples. The tensile testing was conducted for both loading and unloading with 200 load increments to measure acoustoelastic coefficients as the parent material for calibrating the ultrasonic method. The data and results have certain significance for the study of material residual stress, but the following comments require further clarification:
1. The CM was used to verify the phased array ultrasonic results. The curves in Figs 11 and 17 should be compared in one figure to make the verification results more intuitive.
2. The RS measurement value of each element is shown in Fig. 16 with some outliers. It is hoped that the authors will provide detailed interpretations of the experimental phenomenon.
3. It is mentioned in the paper that the sample is asymmetrical and has some slight distortion during the production process. Whether this phenomenon interfere with the stress distribution state of the sample and the TOF measurement results with the phased array probe?

Reviewer 2 Report
Comments and Suggestions for Authors
Please find attached my comments in the docx file.

Reviewer 3 Report
Comments and Suggestions for Authors
In this manuscript, the authors investigated the application of phased array ultrasonics testing (PAUT) on the detection of residual stress on wire arc additive manufacturing methods. By comparing the PAUT with the conventional contour method, the authors verified the reliability of the PAUT with improved accuracy and the enlarged dataset. The provided analysis and discussion make sense and the obtained results support PAUT as a powerful tool to characterize the residual stress of the workpiece, which is quite interesting to the broad readership of this journal. The manuscript is well-organized overall. I just have a few comments on correcting some format issues.
1. In line 135, a period is missed between CM and Ahmad.
2. LCR should be LCR.
3. In line 374, “where” should not be italicized.
4. In line 384, “represents stress variation and” should not be italicized.
5. The introduction is lengthy. Please make it clear and concise.
Based on the abovementioned comments, this manuscript is recommended for minor revision. A revised manuscript is required.
Reviewer 4 Report
Comments and Suggestions for Authors
1. The article is formatted without following the requirements for manuscript formatting in the publishing house.
2. The introduction is presented in great detail and resembles a dissertation or student paper review. It is necessary to briefly outline the relevance, provide a brief critical analysis of the works on the research problem, indicate the scientific novelty and the purpose of the research.
Round 2
Reviewer 4 Report
Comments and Suggestions for Authors
I recommend accepting